# Semi-Automatic Volumetric and Standard Three-Dimensional Measurements for Primary Tumor Evaluation and Response to Treatment Assessment in Pediatric Rhabdomyosarcoma

**DOI:** 10.3390/jpm11080717

**Published:** 2021-07-26

**Authors:** Ewelina Gowin, Katarzyna Jończyk-Potoczna, Patrycja Sosnowska-Sienkiewicz, Anna Belen Larque, Paweł Kurzawa, Danuta Januszkiewicz-Lewandowska

**Affiliations:** 1Department of Health Promotion, Poznan University of Medical Sciences, Święcickiego 6 Street, 60-781 Poznan, Poland; ewego@ump.edu.pl; 2Department of Pediatric Radiology, Poznan University of Medical Sciences, Szpitalna Street 27/33, 60-572 Poznan, Poland; jonczyk@ump.edu.pl; 3Department of Pediatric Surgery, Traumatology and Urology, Poznan University of Medical Sciences, Szpitalna Street 27/33, 60-572 Poznan, Poland; 4Department of Pathology, Hospital Clínic, Villarroel, 170, 08036 Barcelona, Spain; anabelen@inate.es; 5Department of Pathology, Hospital of Lord’s Transfiguration, University of Medical Sciences, Długa Street 1/2, 61-848 Poznan, Poland; paul.kurzawa@yahoo.com; 6Department of Pediatric Oncology, Hematology and Transplantology, Poznan University of Medical Sciences, Szpitalna 27/33, 60-572 Poznan, Poland; danuta.januszkiewicz@ump.edu.pl; 7Department of Medical Diagnostics, Dobra 38a, 60-595 Poznan, Poland

**Keywords:** child, magnetic resonance, solid tumor, treatment response, tumor volume

## Abstract

Current prognostic classification of rhabdomyosarcoma in children requires precise measurements of the tumor. The purpose of the study was to compare the standard three-dimensional (3D) measurements with semi-automatic tumor volume measurement method concerning assessment of the primary tumor size and the degree of response to treatment for rhabdomyosarcoma in children. Magnetic Resonance Imaging data on 31 children with treated rhabdomyosarcoma based on the Cooperative Weichteilsarkom Studiengruppe (CWS) guidance was evaluated. Tumor sizes were measured by two methods: 3D standard measurements and semi-automatic tumor volume measurement (VOI) at diagnosis, and after 9 and 17/18 weeks of the induction chemotherapy. Response to treatment and prediction values were assessed. The tumor volume medians calculated using VOI were significantly higher in comparison with those calculated using the 3D method both during the diagnosis as well as after 9 weeks of the chemotherapy and during the 17–18th week of the treatment. The volume measurements based on the generalized estimating equations on the VOI method were significantly better than the 3D method (*p* = 0.037). The volumetric measurements alone can hardly be considered an unequivocal marker used to make decisions on modification of the therapy in patients with rhabdomyosarcoma.

## 1. Introduction

Rhabdomyosarcoma (RMS) is the most common malignant solid tumor at the developmental age after neuroblastoma and nephroblastoma (Wilms tumor) [1,2,3]. RMS accounts for 5% to 10% of all pediatric neoplasms [4]. Despite the significant progress in the treatment of RMS in children, treatment failure is still observed in a part of them [1]. The determination of the tumor recurrence risk group is of key importance for the selection of the applied therapy. For this purpose, the stage of disease defined before the treatment (Tumor Nodes Metastasis (TNM) Pre-treatment Staging Classification), belonging to the surgical-pathologic group (Intergroup Rhabdomyosarcoma Studies (IRS) Clinical Group Classification), and histological type of the neoplasm are taken into account. The neoplasm stage before the treatment depends on the tumor localization, the extent of infiltration, lymph node involvement, and presence of distant metastases [1,4]. However, the surgical–pathological group results from the level of the tumor resection completeness. Current prognostic classification of RMS in children (according to Cooperative Weichteilsarkom Studiengruppe (CWS)-guidance Version 1.6.1. from 24 May 2014) combines all of the above-mentioned classifications. This classification requires precise measurements of the tumor and the assessment of the response to the used treatment, including the determination of percentage regression or increase in the tumor volume. These measurements induce defined therapeutic interventions directed towards both the intensification and the termination of treatment [4,5].

A more precise determination of the tumor size and volume is necessary not only for the diagnosis but, first and foremost, for the monitoring of the applied treatment. The volumetric measurements of the tumor are becoming more and more important for the evaluation of the clinical stage, assessment of response to the treatment, and also to determine the eligibility for further registration in the clinical studies [6]. The analysis of the tumor size reduction is still one of the most important elements of the therapeutic intervention plan [7,8,9].

Currently, estimation of the tumor size is based on the two-dimensional measurements (2D) using Response Evaluation Criteria In Solid Tumors (RECIST) standards [10,11]. Despite the fact that RECIST criteria recommend the utilization of 1D and 2D measurement for the determination of the therapeutic response and quantitative assessment of the disease progression, Eisenhaue et al. demonstrated that there is a significant non-conformity between measurements by the 1D or 2D method in comparison with the so-called three-dimensional volumetric assessment (3D) [6]. Automatic calculation of the tumor volume based on the 3D measurements is justified only in the case of regular lesions of spherical or ellipsoidal shape. The measurement techniques for these “partial volumes” in case of tumors of an irregular shape differ significantly from the tumor volumetric measurement results obtained using automatic volume measurement. This last type of the volume measurement utilizes the interpolation method which requires manual initiation. Subsequently, the universal imaging software platforms for 3D reading and advanced visualization are utilized to obtain the quantitative measurement of the volume for the selected area. The main limitation of this method is its time consumption and required involvement of an experienced radiologist [12]. The measurement is not accepted as fully reliable if the manual selection is inaccurate and crosses the tumor boundaries. An immense advantage of the semi-automatic volumetric measurement is the fact that the method as the only one enables a complete volume determination of a tumor of an irregular shape and of lesions with multifocal structure.

The purpose of the study was to compare the standard 3D measurements with the semi-automatic tumor volume measurement method concerning the assessment of the primary tumor size and the degree of response to treatment for rhabdomyosarcoma in children.

## 2. Materials and Methods

### 2.1. Patients

The study group consisted of the children diagnosed with soft tissue tumors and treated at Karol Jonscher University Hospital of Karol Marcinkowski University of Medical Sciences in Poznan during the 2006‒2017 period. The parents of the patients have provided informed consent. Consent of the bioethics committee was waived because of the retrospective and non-invasive nature of the study. These rules are compliant with the guidelines of the Bioethical Commission of Poznan University of Medical Science.

This research did not receive any specific grant from funding agencies in the public, commercial, or not-for-profit sectors.

In the group of 57 children diagnosed with soft tissue tumors, 31 children diagnosed with RMS, including 23 with ERMS (embryonal type of RMS) and 8 with ARMS (alveolar type of RMS), were accepted into the study. The children were treated based on the CWS-guidance (Cooperative Weichteilsarkom Studiengruppe (CWS) of the Gesellschaft für Pädiatrische Onkologie und Hämatologie (GPOH)) according to the risk groups [4,5].

The inclusion criteria for the study included: age (up to 17 years old), type of tumor (histopatologically confirmed RMS tumor), treatment in accordance with the adopted schemes, access to the results of the same diagnostic tests, parental consent upon admission to treatment at a university center. The exclusion criteria included: patients who started treatment in another center, children who, for various reasons, did not follow the therapy regimen we adopted, children whose diagnostic tests made it impossible to assess using the above-mentioned methods (incomplete diagnostics, poor quality of imaging tests).

### 2.2. Magnetic Resonance Imaging (MRI)

The tumor evaluation was performed using MRI with resolution below 1 mm of pixel size in each direction and below slice thickness of 4 mm. Recommended MRI-sequences included: pre-contrast T2 weighted sequences with fat suppression, which should be performed in at least two planes of sectioning including axial section. Fat-suppressed T2 weighted images (STIR) will provide necessary information about lymph nodes; dynamic scanning following the contrast is recommended; post-contrast T1 weighted images with fat suppression in at least two planes (mandatory; alternatively 3D-sequence with fat suppression).

### 2.3. Tumour Volume Measurement

#### 2.3.1. Method 1—3D

Tumor dimensions were recorded in three diameters, based on the choice of three maximum widths. Tumor volume (V) calculation for the ellipsoid-shaped tumors was performed manually, when a = length (in cm), b = width (in cm), c = thickness (in cm). V = p/6 * a * b * c = 0.524 * a * b * c in cm^3^.

#### 2.3.2. Method 2—Volume of Interest (VOI)

The measurement of the tumor volume was conducted using an interpolation method. VOI Freehand apparatus method available on the Syngo.via diagnostic stations (Siemens Healthcare) was used for semi-automatic volumetric measurement. The measurement was conducted using transverse plane scans which were the most suitable for the determination of tumor boundaries. The MRI examinations in the study were analyzed using the Magneton Spectra 3T (2013, Siemens Healthcare Germany, Duisburg, Germany) apparatus with a diagnostic station containing Syngo.via software, which enabled the assessment of measurements from various research centers. The detailed analysis of the MRI imaging was based on manual selection of the area of interest on the first and last target plane and on several middle planes. As soon as the measurements are accepted by the operating radiologist, the program calculates the VOI giving the tumor volume measurement in cm^3^. If the tumor was too irregular, it was divided into the respective fragments, the measurements were conducted individually and subsequently summed up. The assessment using the semi-automatic method was conducted 10 times to increase the accuracy and reliability of the tumor volumetric measurements. The measurement standard error was estimated. In the case of the measurement of several tumor components, the measurement standard error was calculated for each area, e.g., a and b. The standard error for the summed value was calculated based on the formula $\sqrt{a^{2}+b^{2}}$.

Additionally, the response to treatment was assessed as the relative reduction (RR%) of the tumor volume. RR in tumor volume after chemotherapy was determined according to the following formula:$$\mathrm{RR}\%=\frac{\mathrm{pretreatment}\;\mathrm{value}-\mathrm{posttreatment}\;\mathrm{value}}{\mathrm{pretreatment}\;\mathrm{value}}*100\%$$
where “value” relates to the tumor volume calculated before and after 9 and 17/18 weeks of the induction chemotherapy. The influence of the tumor reduction after treatment on the results of the therapy were analyzed using a logistic regression model for measurements utilizing the 3D and VOI method. Following tumor reduction cut-off values were accepted: 33%, 50%, and 66%; the status was coded as 1—presence of remission, 0—no remission.

The statistical analysis was performed using Wilcoxon signed-rank test with *p* < 0.05 significance level. Generalized estimating equations were used for the assessment of the prediction value for the volumetric tumor measurements in time. The tumor size and measurement times were the independent values, and the dependent value was the status coded as 1—remission, 0—no remission.

### 2.4. Response Evaluation Criteria

Subsequently, the reduction of the tumor volume between the measurement at diagnosis and after the third week of the chemotherapy cycle (ninth week of the treatment) was analyzed. Based on the CWS protocol, the type of the response to the utilized treatment at this stage determines definite therapeutic procedure. In the case of no regression of 33% in tumor volume, the patients qualify for second-line therapy. The influence of the tumor reduction after 9 weeks of treatment as well as after 17–18 weeks of treatment on the probability of patient survival was also analyzed with the use of logistic regression model for the measurements using the 3D method and the VOI method. The response must last at least 4 weeks without evidence of tumor progression or relapse during this period [4,5].

The overall survival is defined as the time from the initiation of the treatment until death from any cause. The median follow-up time for all patients included in the study was 98 months.

## 3. Results

In the group of 31 children, 20 patients (64%) survived, including two patients in second line therapy due to recurrence. In the studied group, 10 patients (32%) died due to the progress of malignancy, one child died of sepsis. The age of the patients during the onset of the disease was ranging from 4 to 191 months, with the median value of 51 months.

The results of the tumor volumetric measurements conducted using the 3D method and semi-automatic VOI method at diagnosis, after 9 weeks (after third cycle of chemotherapy) and in 17–18th week of treatment are present in Table 1.

In the case of 9 out of 31 children (29%) the semi-automatic method covered the sum of several components due to the irregularity or multifocality of the malignancy. The tumor volume median values calculated using the semi-automatic method—VOI were significantly higher in comparison with those calculated using the 3D method both during the diagnosis as well as after 9 weeks of the chemotherapy and during 17–18th week of the treatment.

Examples of tumors subjected to volumetric assessment are presented in Figure 1 and Figure 2.

Good response to the utilized chemotherapy, assessed as reduction of the tumor size by 33%, was observed in 30/31 of patients (97%) in case of the 3D method, but only in 26/31 children (84%) in the case of the VOI method. The difference, however, was not of statistical significance. The median value of RR% coefficient for the whole tested group using the 3D measurement method was 83.9 (mean and SD = 111.3 ± 180.2) and for the measurement using the VOI method 101.2, respectively (mean and SD = 213.1 ± 388.4). Comparison of RR% values demonstrated that the measurement of the tumor volume reduction using VOI method was not significantly different in comparison with 3D method (*p* = 0.4). The detailed analysis of the response to initial 9-week chemotherapy according to CWS criteria for the 3D and VOI methods is shown in Table 2.

There is no influence observed of the tumor volume reduction by a defined percentage (33%, 50% and 66%) concerning both measurement methods, both in the ninth week as well as in 17–18th week of the treatment.

The prediction value of the tumor volume measurements in correlation with patient survival was additionally analyzed on the basis of the generalized estimating equations. This analysis showed that in case of the VOI method the influence of measurement is statistically significant (*p* = 0.037) in comparison with the 3D method where only borderline significance was observed (*p* = 0.079).

## 4. Discussion

Standard methods of assessing tumor response to applied treatment, especially for metastatic or irregularly shaped tumors, are not entirely reliable. New methods are being introduced such as VOI, or diffusion-weighted magnetic resonance imaging (DWI) and diffusion weighted imaging with background body signal suppression (DWIBS). All of them are very promising techniques for analyzing tumor changes during treatment.

The conducted analysis showed significant differences between the volumes measured using the semi-automatic VOI method in comparison with the 3D method. The volume measured using the 3D method was significantly lower in comparison with the volume measured using the semi-automatic VOI method. Similar results were obtained in studies by Meier et al. and Ziegele et al. [13,14]. The differences between the used methods were due to tumor irregularities which lead to errors in the measurements for each of the three dimensions. This error in the 3D method can be decreased by modification of the volume calculation method, for example by the implementation of the semi-automatic VOI method. Furthermore, the manual measurements exhibit high variability and often lack of measurement repeatability. Semi-automatic measurements provide the possibility to correct the above-mentioned errors by the introduction of the correction factors and also to increase the chance of higher repeatability of the measurements [15].

In the case of the assessment of response to treatment, the analysis of tumor regression is of key importance [16,17]. Our study group is unusual in the way that RMS are neoplasms with unfavorable prognoses. The studied group experienced high mortality with deaths both in the case of responding as well as non-responding patients. Most of the studies describing the usefulness of the semi-automatic VOI method relate to neoplasms in adults [18]. The neoplasms in the pediatric population, including rhabdomyosarcoma, exhibit distinct biology, irregularity, and extremely dynamic growth with infiltrations. Hence, the advantage of this study is the demonstration of the usefulness of the semi-automatic tumor volume measurement method in the population of pediatric patients. Comparison of both methods, 3D and VOI, did not show statistically significant differences, therefore both methods indicated the relative reduction of tumor volume. In Lee’s and Yeo’s study the volumetric measurements obtained using the semi-automatic method correlated well with the histopathological regression of the lesions in both cervical and colorectal cancer [16,19,20,21]. The authors emphasize that the type of method is of particular importance in the case of necrotic areas which decrease the actual tumor volume in comparison to the value obtained using the 3D method. The semi-automatic VOI methods enable the separation of necrotic areas and oedema from the tumor mass.

The percentage decrease in the tumor volume was not a useful prognostic tool for the overall survival. The volume measurements based on generalized estimating equations in the VOI method were significantly better than the 3D measurements (*p* = 0.037), but neither of these methods could be used for unequivocal prediction of the overall survival. Similar results were described by Aghighi et al., Ferrari et al. and Vaarwerk B et al. [1,7,22]. These authors did not demonstrate a significant difference between the tumor volume and its diameter as a prognostic factor in rhabdomyosarcoma and Ewing sarcoma, while other authors showed that the tumor volumetric measurements were more sensitive in detection of no response to the therapy [12,13,15,23,24,25,26,27,28,29,30] or were even the prediction marker of survival in neuroblastoma [26].

The limitation of our study is the small population of patients. Most of the children accepted into the studies exhibited good initial response to the treatment. This fact reduced the ability to determine the strong relationship between reduction of tumor size and survival. Other limitations resulting from the same measurement method cover the analytical diversity mentioned above. Study limitations include also the retrospective nature of the study and a single center design.

## 5. Conclusions

The semi-automatic or automatic tumor volume measurement is currently an accepted method for the determination of the size of solid tumors and the response to treatment. However, the reduction of the tumor volume in RMS in itself in the studied group of patients was not an unequivocal prognostic factor. Therefore, other prognostic factors in RMS remain significant, including primary localization of the lesion, the size of tumor necrosis after chemotherapy assessed based on a histopathological examination or molecular markers, which can be used for the determination of the targeted therapy. Although our results suggest that the volume measurements based on the VOI method may be better than the 3D measurements, either of the volumetric measurements alone can hardly be considered an unequivocal marker used to make decisions concerning modification of therapy in patients with rhabdomyosarcoma.

## Figures and Tables

**Figure 1 jpm-11-00717-f001:**
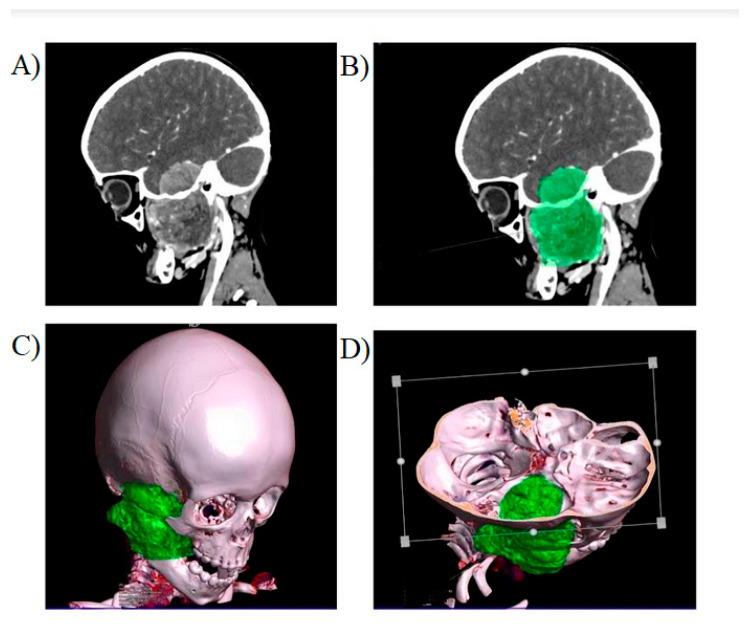
Craniofacial rhabdomyosarcoma tumor. (**A**) Contrast enhancement of the lesion is visible in the sagittal plane. (**B**) The lesion is marked in green for volumetric evaluation. (**C**,**D**) Volume Rendering Technique (VRT) is used for assessment of the tumor.

**Figure 2 jpm-11-00717-f002:**
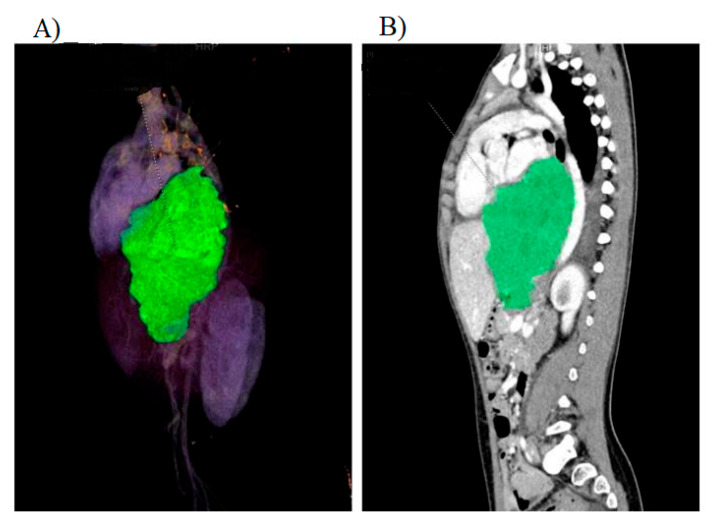
Rhabdomyosarcoma tumor of the mediastinum and abdominal cavity. (**A**) Volume Rendering Technique (VRT) is used for assessment of the lesion. (**B**) Contrast enhancement of the tumor is visible in the sagittal plane.

**Table 1 jpm-11-00717-t001:** Tumor volume calculated according to 3D and semiautomatic VOI method in children with RMS.

	Tumor Volume at Diagnosis (in cm^3^)	Tumor Volume after 9 Weeks of Initial Chemotherapy (in cm^3^)	Tumor Volume after 17–18 Weeks of Treatment (in cm^3^)	
Tumor volume calculation (3D)	2.1–754	0–409.6	0–557.7	Range
	77.3	14.2	0.6	Median
	258.9	143.2	214.5	IQR
	156.9	58.5	51.9	Average
	191.8	106.1	159.6	SD
Semiautomatic tumor volume calculation (VOI)	4.78–1620.22	0–508.1	0–700.1	Range
	112.01	20.98	6.1	Median
	238.9	209.4	269.1	IQR
	156.9	84.1	69.8	Average
	191.8	155.1	199.3	SD
Statistics	3.5385	3.1678	3.0341	Z-value
	0.0004	0.0015	0.0031	*p*-value

**Table 2 jpm-11-00717-t002:** Evaluation of the response to the initial chemotherapy (week 9 of treatment) according to the measurement of tumor volume by 3D and VOI according to the CWS criteria.

Response Evaluation Criteria	3D Method	VOI Method
Complete Response (CR)	0 (0%)	0 (0%)
Very Good Partial Response (VGPR)	5 (16%)	4 (13%)
Partial Response (PR > 2/3)	7 (23%)	5 (16%)
Minor Partial Response (PR < 2/3)	18 (58%)	20 (65%)
Stable Disease (SD)	0 (0%)	2 (6%)
Progressive Disease (PD)	1 (3%)	0 (0%)

## Data Availability

Data available on request due to restrictions.

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
