# Peer review of "Semi-Automatic Volumetric and Standard Three-Dimensional Measurements for Primary Tumor Evaluation and Response to Treatment Assessment in Pediatric Rhabdomyosarcoma"

_jpm, 2021, doi:10.3390/jpm11080717_

Round 1

Reviewer 1 Report

The standard methods for tumor response evaluation, especially in case of tumors with metastases or tumors with irregular shapes, are not really reliable. Novel methods are needed for evaluation. One such possibility could be the application of VOI. The disadvantage of the publication is the low patient number. However the methods are described well and the conclusions are restrained. I have missed the clinical data of the analyzed patients, especially the clinical decisions made upon the 2 different evaluations. 

Questions and suggestions:

  1. On page 5 authors describe the difference of the results of the two methods in 5 cases. Which method was the bases of the clinicians decision? How was the outcome of these patients.
  2. What about the changes in activity of the tumors from the diagnosis, during treatment? How did for example change the DWIBS results?
  3. Only changes in the tumor size effect the treatment decisions? What impact could have changes in activity?
  4. I would recommend to insert a table with clinical outcomes and response evaluation upon the 2 methods.
  5. Did the localization of the tumor have an effect on the results?

In summary it is a well designed, well written paper about an important topic which could support the clinicians decisions. Unfortunately the patient number is quite low, why this topic will need further evaluation.

Author Response

The answer to the Reviewer's Comments:

  1. On page 5 authors describe the difference of the results of the two methods in 5 cases. Which method was the bases of the clinicians decision? How was the outcome of these patients.

Re: We analysed a group of 31 children with rhabdomyosarcoma. This was a retrospective study. Volumetric measurements of the tumour using the 3D method were performed according to the protocol at diagnosis, after the third cycle of chemotherapy i.e. after 9 weeks and at 17-18 weeks of treatment. At the same time intervals, tumour volume was retrospectively assessed using the semi-automated VOI method.

Based on the CWS protocol, the type of the response to the treatment at 9 week determines definite therapeutic procedure. In the case of no regression of 33% in tumor volume, the patients was qualify for second-line therapy.  According to the protocol to evaluate tumour regression we used the measurements using the 3D method.

Good response to the chemotherapy, assessed as reduction of the tumor size by 33%, was observed in 30/31 of patients (97%) in case of the 3D method, but only in 26/31 children (84%) in the case of the VOI method. Among these 5 children, who differed in their response according to the imaging method, four had a relapse ending in non-response to treatment.

  1. What about the changes in activity of the tumors from the diagnosis, during treatment? How did for example change the DWIBS results?

Re: Diffusion-weighted magnetic resonance imaging (DWI) provides information about the local micro structural characteristics of the diffusivity of water molecules in tissues. Diffusion Weighted Imaging with Background body signal Suppression (DWIBS) which stands for diffusion-weighted whole-body imaging with background body signal suppression is an improved DWI technique, offering heavy diffusion weighting, enhanced STIR fat suppression and the possibility of free-breathing. DWIBS is a very promising technique for the analysis of tumour lesions during the course of treatment, but the aim of our study was to compare VOI techniques with classical MR imaging for tumor response evaluation. Reviewer's comment is a very valuable indication for us, which we plan to use in further studies on changes in tumour tissue activity during the course of treatment.

In the discussion of the paper we added the following sentences:

Standard methods of assessing tumour response to applied treatment, especially for metastatic or irregularly shaped tumours, are not entirely reliable. New methods are being introduced such as VOI, or diffusion-weighted magnetic resonance imaging (DWI) and diffusion weighted imaging with background body signal suppression (DWIBS). All of them are very promising techniques for analysing tumour changes during treatment. [Line 232-236]

  1. Only changes in the tumor size effect the treatment decisions? What impact could have changes in activity?

Re: DWIBS MRI sequence is an effective method for detection of solid organ, bone and lymph node metastasis but not specific for characterization of lesions. DWIBS can be used also to elicit metastases.  In some studies DWIBS was 100% sensitive in detection of metastatic bony lesions and had positive predictive value = 100%. So, DWIBS provides a functional assessment of disease, can quantify disease extent. The major advantages of the DWIBS sequence is that it can detect and assess the extra-osseous lesions. The relative availability of this technique and the local experience of radiologists should be considered first. On the other hand, lesions visualised by the DWIBS technique will need to be verified by histopathological analysis, which is not always possible.

  1. I would recommend to insert a table with clinical outcomes and response evaluation upon the 2 methods.

Re: There was no influence observed of the tumor volume reduction by a defined percentage (33%, 50% and 66%) concerning both measurement methods, both in the ninth week as well as in 17-18th week of the treatment. The prediction value of the tumor volume measurements in correlation with patient survival was additionally analysed on the basis of the generalised estimating equations. This analysis showed that in case of the VOI method the influence of measurement is statistically significant (p=0.037) in comparison with the 3D method where only borderline significance was observed (p=0.079). Hence, we saw no purpose in presenting a detailed clinical analysis for both 3D and VOI assessment methods in a separate table.

  1. Did the localization of the tumor have an effect on the results?

Re: The location of soft tissue tumours is mainly the head and neck region, the extremity region, the thoracic region and the pelvic region, including the bladder and genitourinary organs. The location of the tumour is associated with a different response to treatment, influences the method of treatment, including the possibility of using proton therapy, radiotherapy, it also determines the possibility of radical surgery. However, what is the most important, tumour localisation influences the possibility of precise radiological assessment of tumour size and its response to therapy. Hence, the use of VOI techniques for better visualisation of the tumour and response to therapy in locations such as the head and neck region or the pelvis.

The goal of our team is to conduct similar studies on a larger number of patients in the future.

One more time I want to thank for all advices. We introduce corrections into the manuscript.

I hope the manuscript now meets all the requirements.

Yours faithfully,

Patrycja Sosnowska-Sienkiewicz

Danuta Januszkiewicz-Lewandowska

Reviewer 2 Report

The authors compared the standard three-dimensional measurements with the semi-automatic tumor volume measurement method concerning assessment of the primary tumor size and the degree of response to treatment for rhabdomyosarcoma in children.

Overall interesting and well written study with very nice illustrations. However there are several issues that need to be addressed before any favourable decision may be made:

  1. The authors stated that consent of the bioethics committee was waived because of the retrospective and non-invasive nature of the study. According to new GDPR regulations Ethic committee approval is mandatory for all studies (even retrospective). Please provide IRB reference.
  2. Also, the authors stated that the parents of the patients have provided informed consent. If study was retrospective how it is possible that the parents gave informed contents for study which does not exist at the time of treatment. Please clarify.
  3. The authors collected dana till 2017 year. Why the patients from later years were not included in study. Inclusion of the patients from 2018 – 2021 will improve the sample size and quality of the results.
  4. Please include primary / secondary outcomes of the study in methodology.
  5. Please provide clear inclusion / exclusion criteria in methodology.
  6. Please provide a median value of follow-up for all patients included in the study.
  7. Please provide interquartile range (IQR) for all median values in results section and Tables, The authors presented only ranges in Tables.
  8. Under the limitations of the study please include retrospective character of the study as well as single-centre design.
  9. Most of the references are older than five years. Please update!

Author Response

The answer to the Reviewer's Comments:

  1. The authors stated that consent of the bioethics committee was waived because of the retrospective and non-invasive nature of the study. According to new GDPR regulations Ethic committee approval is mandatory for all studies (even retrospective). Please provide IRB reference.

Re: According to the guidelines of the Bioethics Committee of the Poznań University of Medical Sciences, due to the retrospective nature of the study, and additionally due to the lack of impact of the study on the patient, it was not necessary to obtain consent for the conducted research analyses. Nevertheless, we applied to the Commission for confirmation on this issue - a scan of the document is attached.

  1. Also, the authors stated that the parents of the patients have provided informed consent. If study was retrospective how it is possible that the parents gave informed contents for study which does not exist at the time of treatment. Please clarify.

Re: Our hospital is a university centre. Accordingly, parents are informed on admission each time and sign a consent for the potential use of clinical information in scientific studies. The data collected for research are of course done without revealing sensitive data from which the patient can be identified.

  1. The authors collected data till 2017 year. Why the patients from later years were not included in study. Inclusion of the patients from 2018 – 2021 will improve the sample size and quality of the results.

Re: We analysed children with rhabdomyosarcoma diagnosed in years 2006-2017. In order to maintain a 3-year follow-up period, the study group just included patients from those years.

Additionally, which is not important in the merit evaluation, we would like to explain that the results of this study constitute a series of publications in the habilitation proceedings. Therefore, in order to keep the studied material consistent, we limited ourselves to the analysis of a given group of children diagnosed in specific years.

  1. Please include primary / secondary outcomes of the study in methodology.

Re: We do not fully understand the Reviewer. Therefore, we regret to say that it is difficult for us to respond to this comment. The whole methodology of the study was described in detail in the methods section.

  1. Please provide clear inclusion / exclusion criteria in methodology.

Re: The inclusion criteria for the study included: age (up to 17 years old), histopatologically confirmed RMS tumor, treatment in accordance with the adopted schemes, access to the results of the same diagnostic tests, parental consent upon admission to treatment at a university center. The exclusion criteria included: patients who started treatment in another center, children who, for various reasons, did not follow the therapy regimen we adopted, children whose diagnostic tests made it impossible to assess using the above-mentioned methods (incomplete diagnostics, poor quality of imaging tests).

The above information has been completed in the manuscript. [Line 105-112]

  1. Please provide a median value of follow-up for all patients included in the study.

Re: The median follow-up time for all patients included in the study was 98 months.

This sentence has been included in the ‘material and methods’ section. [Line 182-183]

  1. Please provide interquartile range (IQR) for all median values in results section and Tables, The authors presented only ranges in Tables.

Re: In accordance with the Reviewer's comments, we have introduced interquartile range(IQR) for all medians in the results table. [Table 1]

  1. Under the limitations of the study please include retrospective character of the study as well as single-centre design.

Re: The aforementioned limitations have been addressed and supplemented in the manuscript. The sentence below has been inserted into the material section.

Study limitations include the retrospective nature of the study and a single center design. [Line 113-114]

Publication limitations were also noted earlier in the discussion.

The limitation of our study is the small population of patients. Most of the children accepted into the studies exhibited good initial response to the treatment. This fact re-duced the ability to determine the strong relationship between reduction of tumor size and survival. Other limitations resulting from the same measurement method cover the analytical diversity mentioned above. [Line 276-280]

  1. Most of the references are older than five years. Please update!

Re: New references have been added to the publication [2, 3, 8, 9, 20]. The order of the other publications has been adjusted accordingly.

Orsatti G, Morosi C, Giraudo C, et al. Pediatric Rhabdomyosarcomas: Three-Dimensional Radiological Assessments after Induction Chemotherapy Predict Survival Better than One-Dimensional and Two-Dimensional Measurements. Cancers (Basel). 2020;12(12):3808. doi:10.3390/cancers12123808

Saleh MM, Abdelrahman TM, Madney Y, Mohamed G, Shokry AM, Moustafa AF. Multiparametric MRI with diffusion-weighted imaging in predicting response to chemotherapy in cases of osteosarcoma and Ewing's sarcoma. Br J Radiol. 2020;93(1115):20200257. doi:10.1259/bjr.20200257

Gondim Teixeira PA, Biouichi H, Abou Arab W, et al. Evidence-based MR imaging follow-up strategy for desmoid-type fibromatosis. Eur Radiol. 2020;30(2):895-902. doi:10.1007/s00330-019-06404-4

Savadjiev P, Chong J, Dohan A, et al. Image-based biomarkers for solid tumor quantification. Eur Radiol. 2019;29(10):5431-5440. doi:10.1007/s00330-019-06169-w

Kubo T, Furuta T, Johan MP, Adachi N, Ochi M. Percent slope analysis of dynamic magnetic resonance imaging for assessment of chemotherapy response of osteosarcoma or Ewing sarcoma: systematic review and meta-analysis. Skeletal Radiol. 2016;45(9):1235-1242. doi:10.1007/s00256-016-2410-y

One more time I want to thank for all advices. We introduce corrections into the manuscript.

I hope the manuscript now meets all the requirements.

Yours faithfully,

Patrycja Sosnowska-Sienkiewicz

Danuta Januszkiewicz-Lewandowska

Round 2

Reviewer 1 Report

I have got sufficient answers to my questions, and the manuscript was supplemented by my recommendations. New references were added to the paper. I recommend to publish it. 

Reviewer 2 Report

The authors adequately responded to most of my queries. However two queries still remain open:

  1. The study limitations which the authors added in methodology (single center design and retrospective character of the study) should be moved to discussion (under limitations of the study), last paragraph
  2. I asked the authors to add primary and secondary outcomes of the study. The authors stated that they did not adequately understand. This is a very usual part of the methodology in most of the studies. The primary outcomemeasure is the outcome that an investigator considers to be the most important among the many outcomes that are to be examined in the study. Secondary outcomesare outcomes measured in studies of treatment effects that are pre-specified in the protocol as being relevant, but less important than the primary outcomes. I hope that after this explanation, the authors will easily add the required outcomes of treatment of the study.

Author Response

Dear Reviewer,

I wish to resubmit the corrected manuscript entitled “Semi-automatic volumetric and standard three-dimensional measurements for primary tumor evaluation and response to treatment assessment in pediatric rhabdomyosarcoma.” for re-evaluation. We would be very grateful if you reconsidered evaluation of our efforts and potential publication of the manuscript in the revised form.

The answer to the Reviewer Comments:

1) The study limitations which the authors added in methodology (single center design and retrospective character of the study) should be moved to discussion (under limitations of the study), last paragraph.

Re: Thank you very much for this comment. The sentence was moved to the appropriate part of the discussion where all the limitations of the study were summarized. [Line 276-277]

2) I asked the authors to add primary and secondary outcomes of the study. The authors stated that they did not adequately understand. This is a very usual part of the methodology in most of the studies. The primary outcome measure is the outcome that an investigator considers to be the most important among the many outcomes that are to be examined in the study. Secondary outcomes are outcomes measured in studies of treatment effects that are pre-specified in the protocol as being relevant, but less important than the primary outcomes. I hope that after this explanation, the authors will easily add the required outcomes of treatment of the study.

Re: Thank you very much for the explanation. The aim of this study was to compare standard 3D measurements with a semi-automated tumour volume measurement method in the assessment of primary tumour size and treatment response of rhabdomyosarcoma in children.

The primary outcome measure we considered the most important was that volume measurements based on generalised estimating equations in the VOI method were significantly better than the 3D method (p=0.037). 

A secondary outcome measure was that median tumour volumes calculated by the VOI method were significantly higher compared with median volumes calculated by the 3D method both at diagnosis and after 9 weeks of chemotherapy and at week 17-18 of treatment, but volumetric measurements alone can hardly be considered a clear marker for deciding whether to modify therapy in rhabdomyosarcoma patients.

We want to thank for all advices. We introduce corrections into the manuscript.

I hope the manuscript now meets all the requirements.

Yours faithfully,

Patrycja Sosnowska-Sienkiewicz

Danuta Januszkiewicz-Lewandowska